# Mycotoxins’ Prevalence in Food Industry By-Products: A Systematic Review

**DOI:** 10.3390/toxins15040249

**Published:** 2023-03-29

**Authors:** Paloma Lopes, M. Madalena C. Sobral, Guido R. Lopes, Zita E. Martins, Claúdia P. Passos, Sílvia Petronilho, Isabel M. P. L. V. O. Ferreira

**Affiliations:** 1LAQV-REQUIMTE, Department of Chemistry, Campus Universitário de Santiago, University of Aveiro, 3810-193 Aveiro, Portugal; 2LAQV-REQUIMTE, Laboratory of Bromatology and Hydrology, Department of Chemical Sciences, Faculty of Pharmacology, University of Porto, 4050-313 Porto, Portugal; 3Chemistry Research Centre-Vila Real, Department of Chemistry, University of Trás-os-Montes and Alto Douro, Quinta de Prados, 5001-801 Vila Real, Portugal

**Keywords:** by-products contamination, spent, food waste, regulated mycotoxins, emerging mycotoxins

## Abstract

The recovery of biomolecules from food industry by-products is of major relevance for a circular economy strategy. However, by-products’ contamination with mycotoxins represents a drawback for their reliable valorization for food and feed, hampering their application range, especially as food ingredients. Mycotoxin contamination occurs even in dried matrices. There is a need for the implantation of monitoring programs, even for by-products used as animal feed, since very high levels can be reached. This systematic review aims to identify the food by-products that have been studied from 2000 until 2022 (22 years) concerning mycotoxins’ contamination, distribution, and prevalence in those by-products. PRISMA (“Preferred Reporting Items for Systematic Reviews and MetaAnalyses”) protocol was performed via two databases (PubMed and SCOPUS) to summarize the research findings. After the screening and selection process, the full texts of eligible articles (32 studies) were evaluated, and data from 16 studies were considered. A total of 6 by-products were assessed concerning mycotoxin content; these include distiller dried grain with solubles, brewer’s spent grain, brewer’s spent yeast, cocoa shell, grape pomace, and sugar beet pulp. Frequent mycotoxins in these by-products are AFB_1_, OTA, FBs, DON, and ZEA. The high prevalence of contaminated samples, which surpasses the limits established for human consumption, thus limiting their valorization as ingredients in the food industry. Co-contamination is frequent, which can cause synergistic interactions and amplify their toxicity.

## 1. Introduction

Worldwide, innovative strategies to produce zero waste through the reuse of industry by-products, such as distillers’ grains from biofuel and brewers’ spent grains, sugar by-products (from sugarcane, sugar beet molasses, and sugar beet pulp), oilseed cakes, and fruit pulp, have gained increasing attention [1,2]. The population growth, the reduction in land use, and the depletion of renewable resources all force us to consciously use the entirety of food products for the benefit of human nutrition and food security [3]. The recovery of proteins, lipids, carbohydrates, vitamins, minerals, and phenolic compounds can generate economic profit while reducing the environmental implications of mismanagement of waste, as well as improving food security [2,3,4,5]. These strategies are beneficial to the goals of reaching the Sustainable Development Goals and the Zero Hunger Challenge [3]. 

The transition to a circular economy requires the prevention of microbiological and chemical risks to guarantee the physicochemical and microbiological stability of reused food by-products. Sustainable solutions for their management remain a great challenge for several industries, mainly due to their potential contamination by mycotoxins, which are natural toxins produced by diverse of fungi as secondary metabolites. Their chemical structures are diverse and are characterized by a variety of heteroatom-containing functional groups. Aflatoxins (AFs), ochratoxins (OTs), fumonisins (FBs), zearalenone (ZEA), and trichothecenes constitute the main toxins that have been described to potentially occur in a variety of foods, from crops to animal products [6,7,8], with no information on their counterpart by-products. 

Among the mycotoxins known to occur in food products, AFs are the most recognized. These include AFB_1_, AFB_2_, AFG_1_, and AFG_2_, AFB_1_ being the most widely studied [6]. These are mainly produced under temperature conditions of 25 to 30 °C by fungi strains of *Aspergillus flavus* and *A. parasiticus*, although other *Aspergillus* strains and *Penicillium* can also produce AFs. They are classified by the International Agency for Research on Cancer (IARC) as carcinogenic to humans (Group 1), and present hepatotoxic and immunosuppressive properties [9]. *Aspergillus* species are also capable of producing other mycotoxins, namely OTs, at conditions of high-water activity (optimal range of 0.77–0.99 [10]) and temperature (optimal temperature conditions of 20–37 °C [10]), ochratoxin A (OTA) being the most widely studied one. However, OTA can also be produced at temperatures as low as 5 °C, thus exacerbating problems in food contamination even with apparent “appropriate” storage at low temperatures [9,10]. OTs, including OTA, can also be produced by *Aspergillus ochraceus* and by some *Penicillium* spp., namely *P. verrucosum* and *P. nordicum*. Evidence of nephrotoxicity, as well as hepatotoxicity, immunotoxicity, carcinogenic, and teratogenic effects, has been observed in animal studies regarding OTA [9,10]. However, no evidence of these effects has been found in humans, which explains its classification by IARC as a possible human carcinogen (Group 2B) [10].

FBs are another group of naturally occurring mycotoxins produced by the fungi *Fusarium verticillioides* and *F. proliferatum*. FBs are divided in four groups (A, B, C, and P) FB_1_ being the most prevalent. FB_1_ has shown nephron and hepatotoxicity in several species [6,7]; however, a definitive linkage of FB_1_ to human cancer has not been determined [11]. As such, FBs are classified by IARC as possible human carcinogens (Group 2B). *Fusarium* species are also responsible for producing ZEA, which is considered an estrogenic mycotoxin mostly associated with cases of precocious puberty in girls exposed to this mycotoxin, although concerns related to the production of reactive oxygen species (ROS) have also been raised. Trichothecenes are another mycotoxin produced by *Fusarium*, but can also be produced by several other fungi, including *Stachybotrys*, *Myrothecium*, *Trichothecium*, *Trichoderma*, *Cephalosporium*, *Cylindrocarpon*, *Verticimonosporium*, and *Phomopsis*. Deoxynivalenol (DON) stands as the most prevalent trichothecene; however, there is no evidence of its carcinogenicity in humans [7]. 

In recent years, the designation “emerging mycotoxins” has been used to describe certain fungal metabolites, such as beauvericin (BEA) and enniatins (ENNs) [8]. Although not clearly defined, this addresses mycotoxins not regulated by legislation or routinely determined, which have been increasingly quantified in foods [8,11]. BEA was first discovered to be produced by the fungus *Beaveria bassiana,* but is also produced by several *Fusarium* species, such as *F. proliferatum*, *F. subglutinans*, *F. verticillioides*, and *F. oxysporum*. BEA is an ionophore, thus enhancing cell membrane permeability and disrupting normal cell metabolism, and has been shown to be cytotoxic in in vitro studies [11]. ENNs, currently composed of 29 species, are also produced by *Fusarium* species and, like BEA, contain ionophoric properties [11]. In animal studies, both BEA and ENNs have been shown to accumulate in fat-rich tissues (including the liver); however, their toxicity and their carcinogenicity have not yet been confirmed in vivo. The ENNs A1 and B1 are the most frequently detected in food and feed. Mycophenolic acid (MPA) can be produced by *Byssochlamys nivea* and by several *Penicillium* fungi, such as *P. brevicompactum* and *P. carneum*. Although MPA is biologically classified as a mycotoxin, it holds the distinction of being the first purified antibiotic. Currently, MPA is widely used as an immunosuppressive and antirheumatic drug [11]. Nonetheless, a high concentration of this compound in foodstuff has been reported in the past, which is undesirable, even if it possesses overall low toxicity [11]. Another emerging mycotoxin is Roquefortine C (RFC), produced by several species of *Penicillium,* including *P. chrysogenum*, *P. crustosum*, *P. expansum*, *P. griseofulvum*, *P. hirsutum*, *P. hordei*, and *P. melanoconidium,* but most notably by *P. roqueforti*, which is used in cheese production [9]. Gliotoxin (GLT) is produced by several fungal species belonging to genera including *Aspergillus*, *Gliocladium*, *Thermoascus*, and *Penicillium.* GLT was primarily isolated from *Gliocladium fimbriatum*, and a considerable number of immunosuppressive actions have been described for this compound [9]. Patulin is another example of an emerging mycotoxin that can be produced by both *Penicillium* and *Aspergillus* species. It is mostly known for its occurrence in apples, and has known toxicity in plants and animals, but its toxicity towards humans remains unclear [9,12]. However, acute symptoms such as nausea, vomiting, and gastrointestinal discomfort have been reported, as well as chronic genotoxic, neurotoxins, and immunotoxic and teratogenic effects. GLT is included in Group 3 of the IARC, classified as non-carcinogenic to humans [12].

The European Commission has established maximum permitted levels of mycotoxins in different foodstuffs [13] and has provided guidance values concerning maximum mycotoxins in animal feeds, although there are no limits established concerning mycotoxins in food industry by-products. Mycotoxins may be found in by-products due to inappropriate storage conditions, namely, high humidity (non-dried by-products) and high temperature (20–37 °C), which are the optimal conditions for mycotoxin-producing fungal growth [7]. As such, drying is a fundamental step to minimize the risk of mycotoxins. Nonetheless, mycotoxin contamination can also occur in dried by-products, even when preserved at low temperatures. Consequently, mycotoxins can be transferred from the raw material to the final product [11]. The safer use of by-products to obtain added-value products requires data concerning its contamination with mycotoxins. 

The goal of this systematic review is: (i) to summarize which by-products from the processing industry have been studied concerning mycotoxin contamination; (ii) to identify the distribution and prevalence of mycotoxins in those by-products; (iii) to critically discuss the major factors that may contribute to the development of mycotoxins in food by-products, also highlighting the gap in the legislation limits of each mycotoxin, which have been here extrapolated from the corresponding food materials. 

## 2. Results

### 2.1. Literature Search Process

From the 541 records identified, only 313 remained after the title and abstract screening; the remaining 228 articles were duplicates, reviews, conference articles, books, notes, or articles released before 2000 or in a different language than English or Portuguese. Based on the title and abstract, 281 articles were removed, as these studies were not related to the occurrence of mycotoxins in by-products. Thus, the remaining 32 papers proceeded to the full text review. Of those, only 16 articles were related to the quantification of mycotoxins in food industry by-products, specifying the number of samples analyzed and the prevalence of mycotoxins (% positive samples) eligible for data extraction [14,15,16,17,18,19,20,21,22,23,24,25,26,27,28,29]. The PRISMA flow-chart adopted in this systematic review is shown in Figure 1.

### 2.2. Mycotoxins’ Prevalence in By-Products

According to data extracted from the 16 articles obtained by PRISMA, the industry by-products that were found to contain mycotoxins were distiller dried grain with solubles (DDGS), brewer’s spent grain (BSG), brewer’s spent yeast (BSY), cocoa shell (CS), grape pomace (GP), and sugar beet pulp (SBP) [14,15,16,17,18,19,20,21,22,23,24,25,26,27,28,29], as summarized in Figure 2. Due to their rich nutrients’ composition, these by-products have high potential to be used in new product development; however, they can also be optimal for fungal growth.

#### 2.2.1. Distiller Dried Grain with Solubles (DDGS)

DDGS is the main by-product of the distilled ethanol industry (beverage or biofuel), being the ground residue of cereal grains, such as maize, rice, and other grains, that are left over after ethanol production via the grains’ starch [30,31]. DDGS is sold locally in wet form, whereas for transportation across longer distances, DDGS is dried to moisture values below 10%, mainly to reduce its weight. The composition of DDGS depends on the source of the raw material (corn, wheat, and sorghum are the most frequent), production plants, and production procedure. DDGS is known as a rich by-product, since much of its nutritional composition consists of crude (6–10% for maize DDGS [14]) and total carbohydrates (52–57% [31]) and proteins (18–39% for maize DDGS [14,21]), which are not used during the grains’ processing, and thus have high potential to be used in the development of new products [14]. However, the assessment of mycotoxins in DDGS revealed high levels of contamination [14,19,20,21,26] despite its relatively low moisture content (7.1% [14]). Table 1 summarizes the studies concerning the assessment of mycotoxins in DDGS. A variable degree of DDGS contamination was reported. Despite this variability, the most significant content appeared to occur for FB_1_ and FB_2_, DON, ZEA, and BEA, while high variability was observed in AFs. This highlights that the storage conditions of this by-product need to be appropriate, since DDGS is rich in hygroscopic biomolecules, such as polysaccharides (ca. 52–57%, Figure 2); which can, thus, rapidly interact with the humidity of the surrounding environment, leading to the increase in moisture to values that are favorable for mycotoxin spoilage.

The concentrations of AFs and FBs in American maize and sorghum DDGS from ethanol production [21] were accessed, and it was found that only 18.9% of the samples contained no AF-related contamination (Table 1). More than a quarter (28%) of samples showed AFs contamination in the range of 1.0 to 20 µg/kg. The remaining 49% of samples contained AF concentrations below 1.0 µg/kg. Furthermore, in relation to FBs, all samples were shown to be contaminated, but all were observed to be below 5.0 mg/kg FBs, and 94% of samples contained concentrations below 1.0 mg/kg [21]. This widespread mycotoxin contamination among the samples can be correlated to the high protein presence in both types of DDGS (18–38% and 31–38% for maize and sorghum, respectively). The analyses of maize DDGS samples imported from the USA into Saudi Arabia also revealed that at least 1 in 4 DDGS samples were positive for at least one mycotoxin [14]. The three most prevalent mycotoxins were ZEA (35.0% prevalence, ca. 167.6 µg/kg), DON (29%, ca. 3.0 mg/kg), and FBs (25%, ca. 1.0 mg/kg) (Table 1). Maize DDGS, sourced majorly from the USA and Asia, also revealed a high prevalence of mycotoxin contamination, as 92% of the samples were contaminated with two or more mycotoxins. The most prevalent mycotoxins were FBs, which were observed in 91% of samples (1036.0 µg/kg), followed by ZEA and DON, which were observed in 85% (227 µg/kg) and 77% (1755.0 µg/kg) of the analyzed samples, respectively. OTA and AFs were also observed in 25.0% (2.0 µg/kg) and 19.0% (2.0 µg/kg) of the samples [26]. 

Maize DDGS samples from Brazil revealed an even higher degree of mycotoxin contamination, since 98% of all samples were contaminated with at least one mycotoxin [20], but the mycotoxin contamination was lower compared to maize DDGS from other studies [26]. Co-contamination was frequent, with 30% of samples being contaminated with two mycotoxins, and 9% with three or more mycotoxins. Of these, FBs were the most prevalent mycotoxins, with FB_1_ being present in 98.8% of samples (ca. 3.2 µg/kg overall) and FB_2_ in 98% of samples (ca. 1.2 µg/kg) (Table 1). AFB_1_ was the third most prevalent mycotoxin, being present in 32% of samples (ca. 1.5 µg/kg), followed by ZEA, with 17% prevalence (ca. 18.2 µg/kg), and DON, with 13% prevalence (ca. 60 µg/kg) [20]. 

The co-occurrence of Fusarium mycotoxins (FBs, DON, ZEA and BEA) in Thai maize DDGS revealed serious contamination problems. More than half of the samples (51%) were contaminated with all analyzed mycotoxins. FBs and BEA were present in 98% of samples, while ZEA and DON had 81% and 49% prevalence, respectively. Moreover, toxin levels were generally high, with mean levels of 9080.0 µg/kg for FB_1_, 5950.0 µg/kg for FB2, 1160.0 µg/kg for DON, 910.0 µg/kg for ZEA, and 350.0 µg/kg BEA. Furthermore, some registered maximum mycotoxin concentrations, such as those of FB_1_ (143,000.0 µg/kg) and FB_2_ (125,000.0 µg/kg), cause acute toxicological problems in humans, representing a severe concern regarding the use of this by-product [19] (Table 1).

#### 2.2.2. Brewer’s Spent Grain (BSG) and Yeast (BSY)

BSG and BSY are two by-products of the brewing industry. BSG is the main by-product, comprising 85% of the by-products generated. Despite having a rich composition of protein (15–30%) and carbohydrates (up to 73%), as represented in Figure 2 [32,33], BSG is undervalued due to its high moisture content (69–75% [22]), which burdens brewers in relation to its storage and transport [32]. On the other hand, BSY corresponds to the yeast surplus [22], and is known to be rich in proteins (45–60%), carbohydrates (12–27%), and ash (6–14%). It also possesses relatively low moisture levels (7–8%) when compared to the BSG (Figure 2) [34]. Table 2 summarizes the studies concerning the assessment of mycotoxins in brewing by-products (BSG and BSY).

BSY and BSG samples (12 each) from EU breweries were analyzed for their AF content. Only one sample of BSG and one sample of BSY (8.3%) were found to be positive for AF contamination, and only AFB_1_ was found at 0.4 µg/kg and 0.2 µg/kg for BSG and BSY, respectively (Table 2). Therefore, these samples of EU-derived BSG and BSY contained only trace levels of AFs. As such, EU-derived BSG and yeast could be considered for the development of new food products [23]. 

Higher content of AFB_1_ was assessed in BSG samples of Argentinian origin (Table 2): (i) one study quantified AFB_1_ content in fresh BSG samples with an average level of 11.76 µg/kg and 57% prevalence, which increased up to 257.0 µg/kg after 7 days of storage [24]; (ii) a survey on the presence of AFB_1_ in BSG destined for swine feed, which considered 16 samples, revealed that AFB_1_ levels reached concentrations of up to 50.4 µg/kg with a 31.3% prevalence [28]; (iii) another study assessed the presence of AFs, alongside with FBs and ZEA (33 samples), and found that 18% of BSG samples were contaminated with 19–44.5 µg/kg of AFB_1_, while 100% of the samples were contaminated with 104.0–145.0 µg/kg of FBs. No ZEA, nor other AFs, apart from AFB_1_, were detected in those samples [29].

A survey on the presence of FBs in BSG from Brazil destined for inclusion in dairy cattle feed revealed that ca. 72.5% of all samples (total of 80) were contaminated with FBs, with average levels of 227.0 µg/kg [22]. Ten samples of Brazilian BSG and BSY were also evaluated for their DON and ZEA contents—BSG samples presented an average of 1068.0 µg/kg, and DON and ZEA showed an average of 1429.0 µg/kg, respectively, while BSY contained ca. 166.0 µg/kg of DON and no ZEA contamination [18]. On the other hand, eleven samples of Brazilian malt BSG were also studied regarding the concentration of OTA, but no contamination was found [17].

#### 2.2.3. Cocoa Shell (CS)

The making of chocolate is a complex process, comprising multiple precise steps to ensure the highest quality. Thus, a high quantity of co- and by-products of chocolate are generated along the way [16,35]. However, as opposed to other chocolate coproducts, such as cocoa butter and nibs, the cocoa shell (CS), which is removed from the beans after roasting, is unused during the rest of the chocolate production process. As such, it is considered a lowly valued by-product [36,37]. However, CS has been shown to be a rich source in carbohydrates (50–61%) and protein (12–18%) (Figure 2) [36], among other compounds, and as such, it poses great potential to be used in new food product development [36,37].

A total of 19 samples of cocoa shell by-product were analyzed regarding their mycotoxin content (Table 3). The results showed 100% prevalence of AFB_1_ and OTA on those samples, but the content was lower than 2.0 μg/kg [15,16]. Although moisture content was not reported in such studies, CS is known for having a moisture content in the range of 4–6% [36,38,39] (Figure 2), but improper storage (in high humidity conditions, for example) might explain the prevalence of mycotoxins which was observed, since various compounds in its composition are hygroscopic, similarly to carbohydrates.

#### 2.2.4. Grape Pomace (GP)

GP stands as the wine-making industries’ main by-product, representing 20–30% of waste generated during the winemaking process in the form of skin, some pulp, stalks, and grape seeds. It is a by-product rich in carbohydrates (12–41%), protein (4–14%), and phenolics (0.3–9%) [40]. As such, there is great value in the use of grape pomace to fortify foods with beneficial carbohydrates and phenolics. However, currently, it is only used in low-value applications, such as animal feeds, due to its high moisture content (58–82%, Figure 2 [41,42,43,44]). On the other hand, the fortification of foodstuff continues to occur on the laboratory scale [40].

The occurrence of OTA in GP was determined in various samples (Red, White, Porto, Moscatel, and mixture for sparkling wine) from different Douro geographical regions in Portugal (Baixo and Cima Corgo and Douro Superior). OTA was detected in 12 of the 13 analyzed samples, with a mean concentration of 0.07 µg/kg. Although the prevalence was high, trace levels of OTA were quantified (Table 3) [25].

#### 2.2.5. Sugar Beet Pulp (SBP)

In the last few years, sugar beets have risen as an alternative to sugarcane in the production of sugar. Thus, SBP constitutes a sliced sugar-depleted sugar beet by-product which is rich in carbohydrates (up to ca. 83% [45,46]) and protein (7–15% [45,47]), as represented in Figure 2 [47]. However, it also possesses a high moisture content (ca. 72–81% (*w*/*w*) [27,48]), even after pressing operations (18–30% dry matter [27,47]). Thus, it becomes a burden to store and properly use, and, thus, rarely has applications beyond local animal feeding [47]. However, if drying steps are employed, SBP can reach 10% moisture [49].

The presence of various mycotoxins was studied in SBP, a major by-product of the sugar industry (Table 3). Forty samples from five regions in France were analyzed for their content of AFB_1_, OTA, DON, ZEA, patulin (PTL), mycophenolic (MPA), roquefortine C (RQC), gliotoxin (GLT), and penicillic acid (PEA). The results showed that only 8 out of 40 samples were found to be positive, thus indicating that 80% of samples were free from any mycotoxin contamination. In the contaminated samples, ZEA and MPA were the most prevalent. ZEA was found in 3/40 samples at concentrations of 1023, 4862, and 6916 µg/kg, whereas MPA was found in 5/40 samples (up to 1436 µg/kg). OTA was detected in 1 sample at 15 µg/kg. RFC was also detected, but at low levels (Table 3) [27].

## 3. Global Data Analysis

The distribution of mycotoxins assessed in the six food by-products identified in this systematic review (DDGS, BSG, BSY, CS, GP, and SBP) is presented in Figure 3, which summarizes the prevalence of mycotoxins (heatmap) and the number of samples included in the studies according to mycotoxin and by-product (scatterplots). DDGS is the most studied by-product in terms of mycotoxin contamination (>600 samples were assessed), followed by BSG (ca. 200 samples), which demonstrates the robustness of these analyses. On the other hand, the number of samples analyzed for BSY, CS, GP, and BSP is very low, thus limiting the comparison between mycotoxins and their prevalence in the six studied by-products. 

Until now, no legislation existed regarding the mycotoxin content in food by-products. Thus, in this systematic review, an approach using the limits established by legislation in foodstuffs (Figure 4) allowed for the discussion to be extrapolated according to the correspondent by-product. EU legislation has established maximum limits of 5 μg/kg for AFB_1_ and 10 μg/kg for the sum of AFs B_1_ + B_2_ + G_1_ + G_2_ for maize to be subjected to sorting or other physical treatment before human consumption, or for use as an ingredient in foodstuffs. For feed materials, guidelines accept up to 20 μg/kg AFB_1_, although feed materials for dairy and young animals should also comply with the limit of 5 μg/kg for AFB_1_. The limit of OTA in unprocessed cereals is also 5 μg/kg, while the limits for FB_1_ + FB_2_, DON, and ZEA in unprocessed maize are 2000 μg/kg, 1750 μg/kg, and 200 μg/kg, respectively. The limits for DON and ZEA in barley are set at 1250 µg/kg and 100 µg/kg, respectively [14]. For feed materials, guidelines accept up to 250 μg/kg, 60,000 μg/kg, 12,000 μg/kg, and 3000 μg/kg for OTA, FB1 + FB2, DON, and ZEA, respectively [26]. The average and maximum contents of mycotoxins legislated in the EU, which were quantified in the six by-products (DDGS, BSG, BSY, CS, GP, and SBP) are summarized in box plot graphics, presented in Figure 4. The lines indicate the maximum legislated levels for foods and the maximum recommended levels for feed purposes. The data extracted from this systematic review indicate that in most of the samples which were assessed, the mycotoxin content exceeded the limits established for materials to be used for human purposes, although for most of them, the average content was within the guidelines for feed materials. It must be highlighted that mycotoxins were found not only in by-products that presented high humidity, but also in those with lower humidity, such as DDGS, CS, and BSY (Figure 2). Based on this information, there is a need for implementation of a risk assessment program. Maximum residual levels of mycotoxins should be set, even for by-products used as animal feed, since very high levels can be reached. Taking into consideration that by-products of the food industry can be involved in a practical solution for animal feed and can counteract the rising prices of feedstuffs and feed, monitoring the mycotoxin content of by-products prior to their inclusion in the diets of animals is crucial, since the quality of animal feed reflects the quality of cattle products. 

## 4. Conclusions

Nowadays, huge quantities of diverse food industry by-products are discarded. However, this systematic review highlights the fact that the scientific information regarding mycotoxin contamination in by-products of the food industry is scarce. Only 16 articles were found concerning the quantification of mycotoxins in six types of food industry by-products. This selection of articles was based on a transparent, accurate, and replicable methodology. Nevertheless, the global data analysis was based on retrospective data up to 2022; thus, the quality of the data was dependent on the original studies, which may have differed from one study to another. Thus, the data may have been biased. Keeping these limitations in mind, it was concluded that: (i) the mycotoxins commonly occurring in food industry by-products are AFB1, OTA, FBs, DON, and ZEA; (ii) the six by-products which were assessed have a high prevalence of contaminated samples which, in most cases, surpass the limits established for human consumption, which is a drawback for their reliable valorization and may hamper their range of applications as ingredients in the food industry; (iii) the implementation of strict, efficient strategies that reduce mold growth, as well as and hygienic precautions against mycotoxins during by-products storage, are needed. This is because co-contamination is frequent and represents an additional problem since the combined adverse effects of those mixtures of contaminants need to be properly studied. Taking into consideration that mycotoxins were found both in by-products that presented high humidity and in those with lower humidity, and that a limited number of studies were considered, the assessment of mycotoxins in other by-products from the processing industry should be promoted, as this information is relevant for their safe reuse.

## 5. Materials and Methods

### 5.1. Search Strategy

The methodology which we applied was the Preferred Reporting Items for Systematic Reviews and Meta-Analyses (PRISMA). We performed a search of publications in the PubMed and SCOPUS databases, using the following keywords: “mycotoxins” OR “aflatoxins” OR “ochratoxins” OR “patulin” OR “trichothecenes” AND “by-product” OR “food waste” OR “spent”. The collection of articles published up to 15 March 2022 was included. A total of 541 publications were identified after compiling the databases (Figure 1). The compiled articles were inserted into the EndNote library to remove duplicates. Additionally, publications prior to 2000, review articles, conference papers, books, and articles written in languages other than English or Portuguese were also excluded from the list.

### 5.2. Exclusion Criteria and Results Obtained

Three of the authors of this publication independently revised the title and abstract of each of the three hundred and thirteen articles. Those not related to mycotoxins in food waste or food by-products were removed. The full texts of the remaining 32 papers were reviewed. Articles with no access to the full text, as well as studies without a description of the number of analyzed samples and/or prevalence of mycotoxins, were excluded from the final list (16 articles). Then, the data concerning food by-products and mycotoxins were analyzed, and their prevalence was collected (Figure 1). In all steps, selection disagreements were solved by meeting with the other four authors and making decisions together regarding the inclusion or exclusion of the articles, according to the previously explained criteria.

### 5.3. Data Processing 

Graphical analyses were conducted using the free software R (R version 4.2.2, RStudio team, 2022). The heatmap representation used the “superheat” package [50]. This package provided a platform on which to visualize the heatmap and add scatterplots as response variables. 

## Figures and Tables

**Figure 1 toxins-15-00249-f001:**
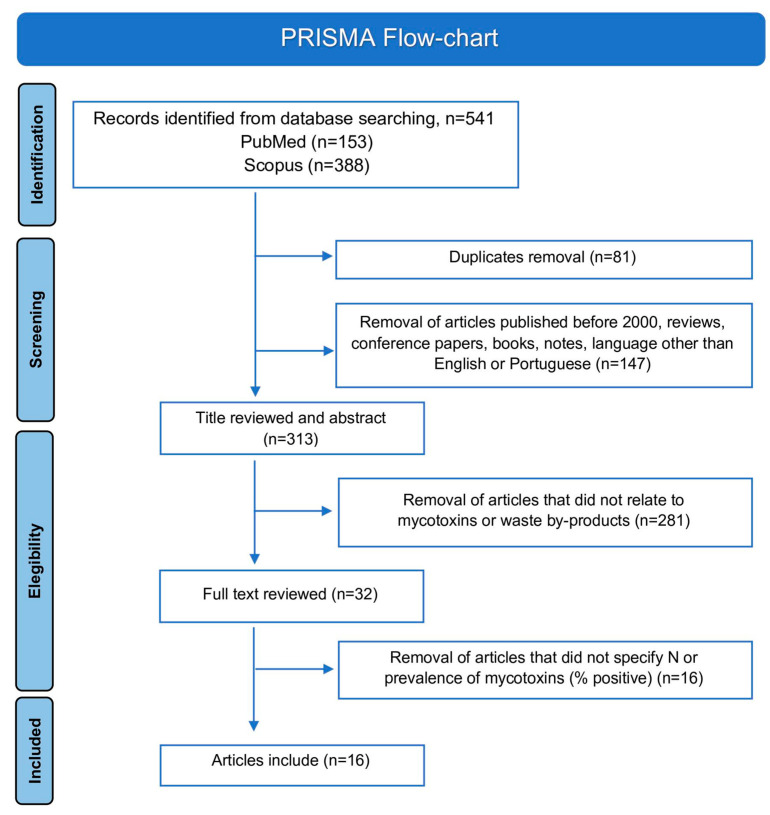
Scheme of the adopted PRISMA flow-chart.

**Figure 2 toxins-15-00249-f002:**
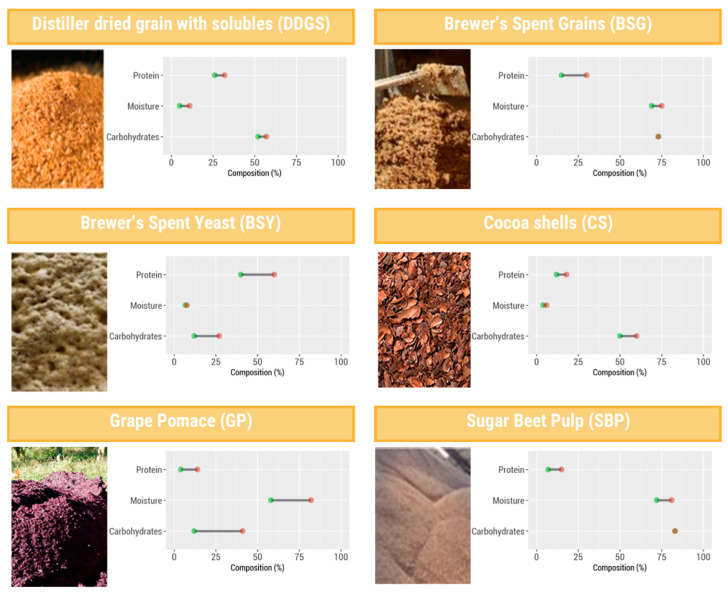
Food industry by-products found to contain mycotoxins, their protein and carbohydrates composition (expressed as % dry weight), and moisture content (%). Green and red dots refer to the minimum and maximum amount reported for each by-product, respectively. Protein, moisture, and carbohydrate data to build the graphics were taken from references [14,21,30,31,32,33,34,35,36,37,38,39,40,41,42,43,44,45,46,47].

**Figure 3 toxins-15-00249-f003:**
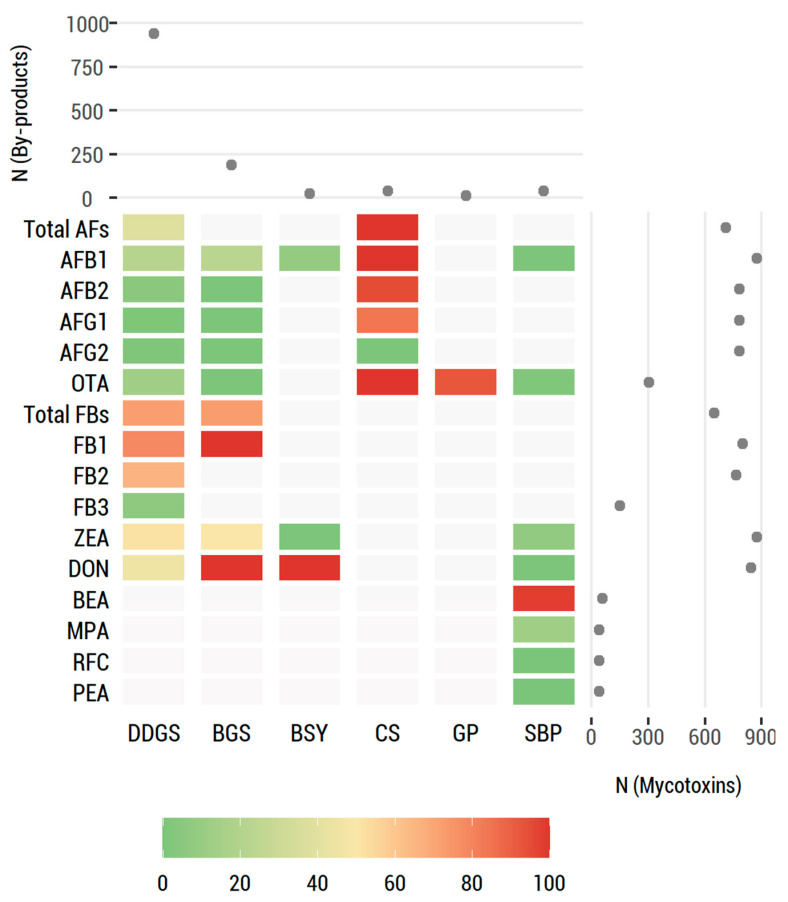
Distribution of mycotoxins studied in food industry by-products, sorted by category (DDGS, BSG, BSY, CS, GP, and SBP). The heatmap summarizes the prevalence of mycotoxins in 6 by-products. Each heatmap cell shows the percentage of positive samples for a given mycotoxin and by-product through a chromatic scale (from dark green (low values) to dark red (high values)). Gray cells correspond to missing values. The right-hand scatterplot displays the number of samples included in the studies by mycotoxin. The upper scatterplot displays the number of samples included in the studies by by-product.

**Figure 4 toxins-15-00249-f004:**
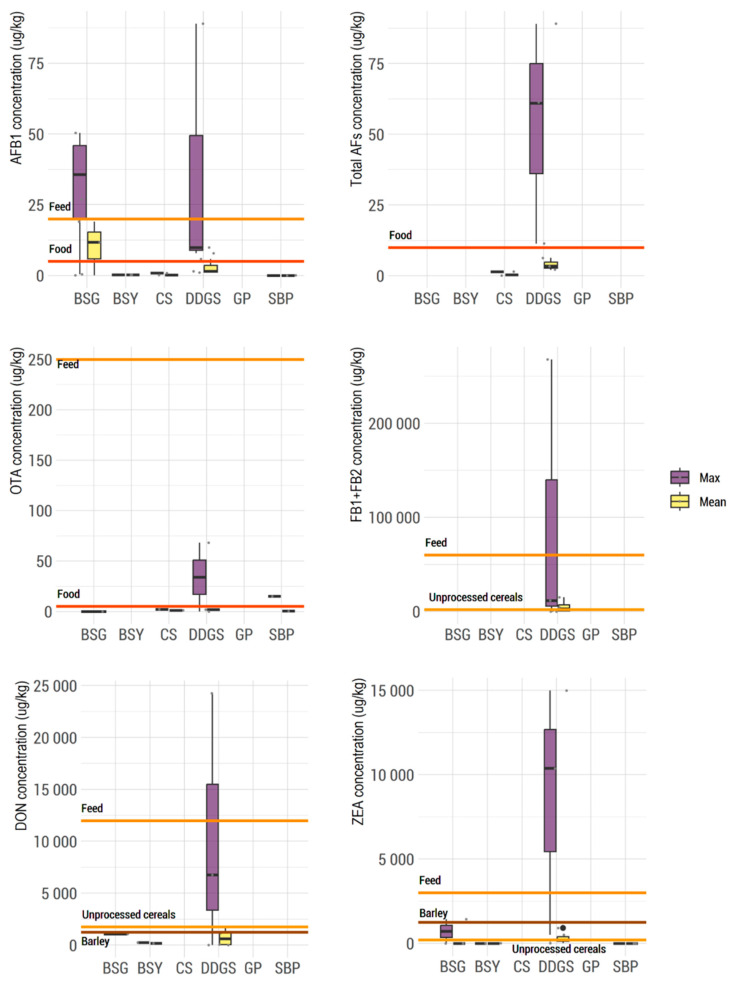
Box plots of the average (yellow) and maximum (purple) contents of mycotoxins legislated in the EU which were found in the 6 food industry by-products (DDGS, BSG, BSY, CS, GP, and SBP). The horizontal lines indicate maximum limits in unprocessed cereals (light brown), barley (dark brown), or foods (red), as well as maximum recommended levels for feed purposes (orange).

**Table 1 toxins-15-00249-t001:** Summary of mycotoxins quantified in DDGS.

Mycotoxin	*n*	% Positive	Maximum (μg/kg)	Average (μg/kg)	Reference
AF Total	150	14.0	11.3	6.3	[14]
	148	81.1	61.0	3.4	[21]
	393	19.0	89.0	2.0	[26]
AFB_1_	150	14.0	9.9	5.8	[14]
	176	32.3	7.9	1.5	[20]
	393	18.0	89.0	1.0	[26]
AFB_2_	150	6.0	0.6	0.5	[14]
	176	3.4	-	0.1	[20]
	393	5.0	19.0	0	[26]
AFG_1_	150	2.7	0.7	0.5	[14]
	176	0	-	-	[20]
	393	1.0	3.0	0	[26]
AFG_2_	150	4.0	1.1	0.8	[14]
	176	0	-	-	[20]
	393	0	0	0	[26]
OTA	47	0	0	-	[20]
	173	25.0	68.0	2.0	[26]
Total FBs	150	25.3	2.2	1..0	[14]
	31	100.0	5.0	0.7	[21]
	390	91.0	9042.0	1036.0	[26]
FB_1_	150	25.3	3.6	1.6	[14]
	59	98.3	143,000.0	9080.0	[19]
	168	98.8	-	3207.0	[20]
	390	91.0	9042.0	892.0	[26]
FB_2_	150	23.3	2.1	0.6	[14]
	59	98.3	125,000.0	5950.0	[19]
	168	97.8	-	1243.0	[20]
	390	44.0	2626.0	144.0	[26]
FB_3_	150	6.0	0.7	0.4	[14]
DON	150	28.7	8.1	3.0	[14]
	59	49.2	6750.0	1160.0	[19]
	168	12.9	-	59.6	[20]
	409	77.0	24,269.0	1755.0	[26]
ZEA	150	34.7	501.0	167.6	[14]
	59	81.4	14,990.0	910.0	[19]
	168	18.0	-	18.2	[20]
	405	85.0	10,374.0	227.0	[26]
BEA	59	98.3	4220.0	350.0	[19]

*n*—number of samples; % of positive samples.

**Table 2 toxins-15-00249-t002:** Summary of mycotoxins quantified in brewing by-products (BSG and BSY).

By-Product	Mycotoxin	*n*	% Positive	Maximum (µg/kg)	Average (µg/kg)	Reference
BSG	AFB_1_	12	8.3	0.4	0.03	[23]
		26	57.0	26.9	11.8	[24]
		16	31.3	50.4	-	[28]
		33	18.0	44.5	19.0	[29]
	AFB_2_	12	0	0	0	[23]
		33	0	0	0	[29]
	AFG_1_	12	0	0	0	[23]
		33	0	0	0	[29]
	AFG_2_	12	0	0	0	[23]
		33	0	0	0	[29]
	OTA	11	0	0	-	[17]
	Total FBs	80	72.5	-	227.0	[22]
	FB_1_	33	100.0	145.0	124.5	[29]
	DON	10	100.0	1068.0	-	[18]
	ZEA	33	0	0	0	[29]
		10	100.0	1429.0	-	[18]
BSY	AFB_1_	12	8.3	0.2	0.02	[23]
	DON	10	100.0	241.0	166.0	[18]
	ZEA	10	0	0	0	[18]

*n*—number of samples; % of positive samples.

**Table 3 toxins-15-00249-t003:** Summary of mycotoxins quantified in cocoa shell (CS), grape pomace (GP), and sugar beet pulp (SBP) by-products.

By-Product	Mycotoxin	*n*	% Positive	Maximum (μg/kg)	Average (μg/kg)	Reference
CS	AFs Total	19	100.0	1.4	0.3	[15]
	AFB1		100.0	0.8	0.2	
	AFB2		95.0	0.02	0.01	
	AFG1		84.0	0.4	0.1	
	AFG2		0	0.06	0.02	
	OTA	19	100.0	2.0	1.1	[16]
GP	OTA	13	92.3	0.1	0.07	[25]
SBP	AFB1	40	0	0	0	[27]
	OTA		2.5	15.0	0.4	
	DON		0	0	0	
	ZEA		7.5	6916.0	71.0	
	PTL		0	0	0	
	MPA		12.5	1436.0	320.0	
	RFC		0	0	0	
	GLT		0	0	0	
	PEA		0	0	0	

*n*—number of samples; % of positive samples.

## Data Availability

Not applicable.

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
