# Peer review of "Mycotoxins’ Prevalence in Food Industry By-Products: A Systematic Review"

_toxins, 2023, doi:10.3390/toxins15040249_

Round 1

Reviewer 1 Report

The manuscript titled " Mycotoxins’ prevalence in food industry byproducts: a systematic review" reviewed the scientific publications related with food byproducts contaminated with mycotoxins from 2000 to 2022.

The manuscript is well-written and thoroughly reviewed the most significant works from the past 22 years. Although the number of articles included is limited (only 16), the review was well-organized and followed the PRISMA guidelines with good criteria. The scope of the review aligns with Toxins, and I recommend it for publication with minor revisions (outlined below).

 - Figure 2. Please improve the quality of the image

-   Figure 4. Please modify the figure the last two graphs have the lines and the text overlapped and it is difficult to read.

-   Tables 1, 2 and 3. Please unify the decimals and the number format of the tables.

Author Response

Comment: The manuscript titled " Mycotoxins’ prevalence in food industry byproducts: a systematic review" reviewed the scientific publications related with food byproducts contaminated with mycotoxins from 2000 to 2022.

The manuscript is well-written and thoroughly reviewed the most significant works from the past 22 years. Although the number of articles included is limited (only 16), the review was well-organized and followed the PRISMA guidelines with good criteria. The scope of the review aligns with Toxins, and I recommend it for publication with minor revisions (outlined below).

Answer: Authors thank the appreciation comments of the review and the suggestions to improve the article.

Comment: Figure 2. Please improve the quality of the image

Answer: The quality of the image was improved.

Comment: Figure 4. Please modify the figure the last two graphs have the lines and the text overlapped and it is difficult to read.

Answer: The figure was modified to assure that the lines and the text does not overlap the lines.

Comment: Tables 1, 2 and 3. Please unify the decimals and the number format of the tables.

Answer: As suggested the decimals and the number format of the tables were unified.

Reviewer 2 Report

The present paper is a systematic overview of the last 2 decades, dealing with mycotoxins contamination, distribution and prevalence in food industry by-products. Overall, the work is interesting and well-written. There are some issues that have to be addressed by authors, for further improvement of the manuscript.

-L5. Please revise “byproducts” to “by-products” here and throughout the manuscript.

-L12-14. Please revise.

-L20. Please revise.

-L22. The last keyword should be removed.

-L32-35. Please split this sentence into 2.

-L37-38. Please revise.

-L43-44. ”namely with mycotoxins” Please revise.

-L130. Which is the meaning of duplicates and why are referred? Please specify.

-L338. I suggest authors to clearly highlight the limitations of the present paper and, as well as to note some “next steps” as future prospects towards this direction.

Author Response

Comment: The present paper is a systematic overview of the last 2 decades, dealing with mycotoxins contamination, distribution, and prevalence in food industry by-products. Overall, the work is interesting and well-written. There are some issues that have to be addressed by the authors, for further improvement of the manuscript.

Answer: Authors thank the appreciation comments of the review and the suggestions to improve the article.

Comment: L5 - Please revise “byproducts” to “by-products” here and throughout the manuscript.

Answer: As suggested “byproduct” was revised to “by-product”.

Comment: L12-14 - Please revise.

Answer: The sentence was revised as follows: “A total of 6 by-products have been assessed concerning mycotoxins content, those include distiller dried grain with solubles, brewer’s spent grain, brewer’s spent yeast, cocoa shell, grape pomace, and sugar beet pulp.

Comment: L20 - Please revise.

Answer: The sentence was revised as follows: “There is a need for the implantation of a constant monitoring program since very high levels can be achieved.”

Comment: L22 - The last keyword should be removed.

Answer: As suggested, the last keyword was removed.

Comment: L32-35 - Please split this sentence into 2.

Answer: The sentence was rewritten as follows: “By the year 2050, it is estimated that the world population will surpass 9 billion [2]. The population growth, the reduction of land use, and the depletion of renewable resources force a conscious use of the entirety of food for the benefit of human nutrition and food security.”

Comment: L37-38 - Please revise.

Answer: The sentence was rewritten as follows: “These strategies are beneficial to reach the Sustainable Development Goals and the Zero Hunger Challenge [2].”

Comment: L43-44 - ”namely with mycotoxins” Please revise.  L47

Answer: namely was removed from the sentence “potential contamination by mycotoxins,”

Comment: L130 - Which is the meaning of duplicates and why are referred? Please specify.

Answer: The literature search was performed in two databases, the PubMed and SCOPUS, using the following keywords: “mycotoxins” OR “aflatoxins” OR “ochratoxins” OR “patulin” OR “trichothecenes” AND "by-product" OR "food waste" OR “spent”. Articles were compiled and the duplicates removed, this is the usual procedure when PRISMA methodology is applied, use at least two different databases and remove the duplicates. Usually, it is detailed in the systematic review to assure that the methodology was correctly applied.

Comment: L338 - I suggest authors to clearly highlight the limitations of the present paper and, as well as to note some “next steps” as future prospects towards this direction.

Answer: Like any review system, systematic reviews have their advantages and limitations. The following information was inserted in the conclusions section: “This selection of articles is based on a transparent, accurate, and replicable methodology. Nevertheless, the global data analysis is based on retrospective data up to 2022, and the quality of the data depends on the original studies, which can differ from one study to another, thus data can have biases. Keeping those limitation on mind,…”

Reviewer 3 Report

Dear Editor,

My review report is attached herewith.

Author Response

Comment: Review report for toxins--2284074 The paper reviews an interesting topic for this journal. Despite this, it needs more improvement before being accepted for publication. The main weakness is the bibliography that should be upgraded by recent reference. Moreover, it should be improved by a native English speaker for editing (for instance “Mycotoxins’” in title is correct?) . Specific recommendations are listed below.

Answer: Authors thank the comments of the review and the suggestions to improve the article. References were updated. English was improved. The title is correct.

Comment: Abstract L.5-7. Re-phrase or delete because it is unclear and confusing.

Answer: The sentence was rewritten as follows: “The recovery of biomolecules from food industry by-products is of major relevance under a circular economy strategy. However, by-products contamination with mycotoxins represents a drawback for their reliable valorisation for food and feed hampering their application range, namely as food ingredients.”

Comment: L.10. Replace acronym PRISMA by “Preferred Reporting Items for Systematic Reviews and MetaAnalyses”.

Answer: PRISMA is an acronym used worldwide, but authors accept the suggestion and inserted the extended designation.  

Comment: L.17-20. This section should be moved at the beginning of the abstract (no at the end).

Answer: As suggested lines 17-20 were moved to the beginning of abstract.

Comment: Key words aflatoxins; ochratoxins; fumonisins; trichothecenes should be merged. Too many words are listed.

Answer: As suggested keywords were improved. The new keywords are: by-products contamination; spent; food waste; regulated mycotoxins;  emerging mycotoxins.

Comment: Introduction Reference [1] should be integrated by other ones.

Please, read the followings (DOI: 10.1016/B978- 0-323-90569-5.00010-X for DDGS and DOI: 10.1016/j.rser.2018.02.041 for other co/by-products).

Answer: Authors read the suggested references:

De Corato, U., De Bari, I., Viola, E., Pugliese, M. Assessing the main opportunities of integrated biorefining from agro-bioenergy co/by-products and agroindustrial residues into high-value added products associated to some emerging markets: A review. Renewable and Sustainable Energy Reviews 88, 2018, 326-346

De Corato, U., Viola, E., “Biofuel co-products for livestock feed' In book: Agricultural Bioeconomy - Innovation and Foresight in the Post-Covid Era, First Edition, Academic Press, Elsevier Inc. 2023, chapter 13, pp.245-286. doi: 10.1016/B978-0-323-90569-5.00010-X.

Both references were very adequate to the introduction section; therefore, they were included as reference [2] and reference [4].

Comment: L.43 “, namely with”. Replace “by”.

Answer: This correct was done in the text.

Comment: L.49 References [6,7] are old. Please, read the following (DOI: 10.3390/toxins15020099) and similar.

Answer: As suggested the reference: Salvatore M.M., Andolfi, A., Nicoletti, R. Mycotoxin Contamination in Hazelnut: Current Status, Analytical Strategies, and Future Prospects. Toxins 2023, 15(2), 99; https://doi.org/10.3390/toxins15020099, was inserted in introduction section ref [8].

Comment: L.80-81. Please, quote the sentence. L.121-126. Please, go to the beginning of the paragraph. Very long and hard sentence to read!! Rewrite it using points at proper places to differentiate the three-focus highlighted by the authors.  

Answer: L80-81 a reference was inserted [8]. L121-126 As suggested the sentence was changed to clearly differentiate the three-focus highlighted by the authors.

Comment: Results L.155. References [30,31] are old. Please, read the following (DOI: 10.1016/B978-0-323-90569- 5.00010-X) and similar (DOI: 10.3390/molecules21030285).

Answer: As suggested both references were inserted in the article: [4] De Corato, U., Viola, E., “Biofuel co-products for livestock feed' In book: Agricultural Bioeconomy - Innovation and Foresight in the Post-Covid Era, First Edition, Academic Press, Elsevier Inc. 2023, chapter 13, pp.245-286. doi: 10.1016/B978-0-323-90569-5.00010-X.

[30] J. Popp, Harangi-Rákos, M., Gabnai, Z., Balogh, P., Antal, G., Bai, A. Biofuels and Their Co-Products as Livestock Feed: Global Economic and Environmental Implications. Molecules 2016, 21(3), 285; https://doi.org/10.3390/molecules21030285

Comment: Global data analyses L.298. “. This figure”. Delete it and merge two-sentences in the only one by “that”.

 L.315. “were presented (Figure 4).”. Move it at the end of the sentence.

Answer: Those sentences were improved.

Comment: Conclusion L.353-361. It seems decontextualized in this section. It should be moved in the proper place of the “Introduction” or “Results” section.

Answer: This information is relevant and was inserted in the Results section, because it was concluded from the analysis of the review data.

Comment: Figure Figure 1. Scheme of the PRISMA flow-chart adopted. Please, quote the caption unless Fig.1 is an original elaboration of the authors.

Answer: Fig 1 is the usual PRISMA flow-chart, it doesn’t need quote.

Comment: Figure 2. The same recommendation as the previous one

Answer: Data to build the graphics was taken from references [14, 21, 31-33, 36, 40, 45-47]

Reviewer 4 Report

Mycotoxins’ prevalence in food industry byproducts: a systematic review - The point and the content is interesting, but plz fix the following points before going to the next step:

- The main text should benifit from a language editing. 

- The figures and tables quality ought to be enhanced. 

- The most importnat, the novelty of this review should be highlighted and compared with other reviews, for example: 

Ünüsan, N. (2019). Systematic review of mycotoxins in food and feeds in Turkey. Food Control97, 1-14.

Khaneghah, A. M., Fakhri, Y., Raeisi, S., Armoon, B., & Sant'Ana, A. S. (2018). Prevalence and concentration of ochratoxin A, zearalenone, deoxynivalenol and total aflatoxin in cereal-based products: A systematic review and meta-analysis. Food and chemical toxicology118, 830-848.

Mihalache, O. A., Dellafiora, L., & Dall'Asta, C. (2022). A systematic review of natural toxins occurrence in plant commodities used for plant-based meat alternatives production. Food Research International, 111490.

Those are just some examples for related references, and thus the authors should compre among. 

- Update the references used and cite the most related ones. 

- What are the critiria you used in the method section and based on what you included or excluded some references, explaine in more detailes.

Author Response

Comment: Mycotoxins’ prevalence in food industry byproducts: a systematic review - The point and the content is interesting, but plz fix the following points before going to the next step:

- The main text should benifit from a language editing. 

- The figures and tables quality ought to be enhanced. 

Answer: Authors thank the appreciation comments of the review. The English was improved and the quality of Figures and tables was improved.

Comment: - The most importnat, the novelty of this review should be highlighted and compared with other reviews, for example: 

Ünüsan, N. (2019). Systematic review of mycotoxins in food and feeds in Turkey. Food Control97, 1-14.

Khaneghah, A. M., Fakhri, Y., Raeisi, S., Armoon, B., & Sant'Ana, A. S. (2018). Prevalence and concentration of ochratoxin A, zearalenone, deoxynivalenol and total aflatoxin in cereal-based products: A systematic review and meta-analysis. Food and chemical toxicology118, 830-848.

Mihalache, O. A., Dellafiora, L., & Dall'Asta, C. (2022). A systematic review of natural toxins occurrence in plant commodities used for plant-based meat alternatives production. Food Research International, 111490.

Those are just some examples for related references, and thus the authors should compre among. 

- Update the references used and cite the most related ones.

Answer: All the interesting reviews mentioned by the reviewer focus on mycotoxins, but none of those reviews focus on mycotoxins on industry by-products, which is the focus and the novelty of this systematic review. This information was highlighted in the conclusion section.

Comment: - What are the critiria you used in the method section and based on what you included or excluded some references, explaine in more detailes.

Answer: This article follows the rules of a systematic review, the Search Strategy is described in sub-section 5.1. and 5.2

Reviewer 5 Report

I positively appreciate the article entitled "Mycotoxins' prevalence in the food industry byproducts: a systematic review".

The abstract reflects the content of the article, is correctly written

The introduction is very comprehensive and gives a well-argued picture of the objectives of this systematic review of mycotoxins in the food industry by-products.

The results are presented, accompanied by a graphical representation, expl. the Flow Chart Prism adopted shows that the data were extracted from 16 articles concerning mycotoxins in by-products of the brewing industry: dried grain with solubles (DDGS), brewer's spent grain (BSG), brewer's spent yeast (BSY), cocoa shell (CS), grape pomace (GP), and sugar beet pulp (SBP.)

 The results are presented in summary, in tabel form, which allows a better overview. 

The analysis of the data on the 6 by-products of the food industry and the results are also presented in the form of a heatmap. 

The approach using the mycotoxin limits set by legislation for food products is interesting and allowed extrapolation to the corresponding by-products. It uses charts that make more visible the average and maximum EU-legislated mycotoxin content of the 6 by-products. 

The conclusions are adequate, as are the materials and methods correctly described. 

Author Response

Comments:

I positively appreciate the article entitled "Mycotoxins' prevalence in the food industry byproducts: a systematic review".

The abstract reflects the content of the article, is correctly written

The introduction is very comprehensive and gives a well-argued picture of the objectives of this systematic review of mycotoxins in the food industry by-products.

The results are presented, accompanied by a graphical representation, expl. the Flow Chart Prism adopted shows that the data were extracted from 16 articles concerning mycotoxins in by-products of the brewing industry: dried grain with solubles (DDGS), brewer's spent grain (BSG), brewer's spent yeast (BSY), cocoa shell (CS), grape pomace (GP), and sugar beet pulp (SBP.)

 The results are presented in summary, in tabel form, which allows a better overview. 

The analysis of the data on the 6 by-products of the food industry and the results are also presented in the form of a heatmap. 

The approach using the mycotoxin limits set by legislation for food products is interesting and allowed extrapolation to the corresponding by-products. It uses charts that make more visible the average and maximum EU-legislated mycotoxin content of the 6 by-products.

The conclusions are adequate, as are the materials and methods correctly described. 

Answer: Authors thank very much the appreciation comments of the reviewer concerning all sections of the manuscript.

Round 2

Reviewer 3 Report

The manuscript has been improved following all recommendations of the reviewers. It Is acceptable for publication.

Reviewer 4 Report

It could be accepted in its current form.